behaviour

parental care, incubation, brood care, nest guarding, *Philornis downsi*, ectoparasite

**Author for correspondence:**
Sonia Kleindorfer
e-mails: sonia.kleindorfer@flinders.edu.au,
sonia.kleindorfer@univie.ac.at

# Female in-nest attendance predicts the number of ectoparasites in Darwin's finch species

Sonia Kleindorfer[1,2], Lauren K. Common[1], Jody A. O'Connor[3], Jefferson Garcia-Loor[2,4], Andrew C. Katsis[1], Rachael Y. Dudaniec[5], Diane Colombelli-Négrel[1] and Nico M. Adreani[2]

[1]College of Science and Engineering, Flinders University, Adelaide 5001, Australia
[2]Konrad Lorenz Research Center for Behavior and Cognition and Department of Behavioral and Cognitive Biology, University of Vienna, Vienna 1090, Austria
[3]Department for Environment and Water, Adelaide 5000, Australia
[4]Charles Darwin Research Station, Galápagos, Ecuador
[5]Department of Biological Sciences, Macquarie University, Sydney 2109, Australia

SK, 0000-0001-5130-3122; NMA, 0000-0002-5043-0389

Selection should act on parental care and favour parental investment decisions that optimize the number of offspring produced. Such predictions have been robustly tested in predation risk contexts, but less is known about alternative functions of parental care under conditions of parasitism. The avian vampire fly (*Philornis downsi*) is a myasis-causing ectoparasite accidentally introduced to the Galápagos Islands, and one of the major mortality causes in Darwin's finch nests. With an 11-year dataset spanning 21 years, we examine the relationship between parental care behaviours and number of fly larvae and pupae in Darwin's finch nests. We do so across three host species (*Camarhynchus parvulus*, *C. pauper*, *Geospiza fuliginosa*) and one hybrid *Camarhynchus* group. Nests with longer female brooding duration (minutes per hour spent sitting on hatchlings to provide warmth) had fewer parasites, and this effect depended on male food delivery to chicks. Neither male age nor number of nest provisioning visits were directly associated with number of parasites. While the causal mechanisms remain unknown, we provide the first empirical study showing that female brooding duration is negatively related to the number of ectoparasites in nests. We predict selection for coordinated host male and female behaviour to reduce gaps in nest attendance, especially under conditions of novel and introduced ectoparasites.

## 1. Introduction

Parental care functions to enhance offspring survival by satisfying offspring needs during growth and development, often at a cost to the parent [1–3]. Selection should act on parental care and favour parental investment decisions that optimize both the number and quality of offspring in relation to the current versus future reproductive opportunity of each parent [4,5]. Such predictions have been robustly tested in the context of predation risk, and parents generally adjust their level of parental defence behaviour towards a predation threat to themselves or their offspring, as predicted by life-history theory [6–9]. In the context of avian nest ectoparasites, the effects of parasitism on parental care have mostly been explored in relation to food delivery. Yet, when offspring are parasitized, research has found inconsistent patterns across species in the impact of parasite burden × chick begging on parental food delivery [10–12]. There are also gaps in knowledge because of missing information about the relative value of current versus future broods infested with ectoparasites,

which further hampers interpretation of adjustments in food delivery to parasitized offspring [13,14]. In this study, we take a different approach and ask a more fundamental question: can parental care at the nest act as a physical deterrent that prevents ectoparasites from entering the nest to oviposit? If longer in-nest attendance by parents deters ectoparasite oviposition, then factors that promote longer in-nest attendance such as brooding duration could be targets of selection when parasites impose strong fitness costs.

Most studies on host parental care in response to ectoparasites have tested the food compensation hypothesis, which predicts that parents will increase food delivery to offspring at parasitized nests to compensate for the nutritional and energetic costs of parasitism [15,16]. Evidence for the food compensation hypothesis has been mixed [17–21], perhaps due to sex differences in the value of current versus future broods [20]. Parental care can reduce the number of ectoparasites when parents consume or remove parasites during nest sanitation [22–24] or use aromatic plants with volatile chemical compounds during nest building or applied to themselves, which can deter ectoparasites [25–27] (but see [28]). However, no studies we are aware of have measured the role of parental in-nest attendance, a possible form of nest guarding, to reduce parasite burden, or the effect of behavioural conspicuousness of the attending birds, measured as parental activity at the nest, to deter or attract ectoparasites.

The offspring of Darwin's finches on the Galápagos Islands are currently being parasitized by the accidentally introduced avian vampire fly (*Philornis downsi*), a myiasis-causing ectoparasite [29]. Adult *P. downsi* were first collected from traps on Santa Cruz Island in 1964 [30]. The adult fly is vegetarian and feeds on decaying plant matter [31], but the developing fly larvae are parasitic and feed on the blood and tissue of avian chicks. These chicks suffer, by far, the most extreme costs of *P. downsi* parasitism, including blood loss, naris deformation, infected body wounds, and mortality [32]. On average, 55% of chicks die in the nest due to parasitism, with annual variation in mortality ranging from 20% to 100% per year [32]. Effects of *P. downsi* larvae on adult birds are considered indirect. For example, *P. downsi*-specific antibodies have been detected in some, but not all, adult finches [12,24], and adult finches that survived parasitism as chicks often sustain enlarged or malformed nares (nasal openings) as adults [33,34]. *Philornis downsi* females oviposit eggs onto nesting material [23] and perhaps onto chicks; after hatching, larvae crawl inside the nares of the chicks to consume blood or keratin [11]. Increasingly over the past decade, larvae on Santa Cruz Island are suspected to consume the blood of incubating females [35], with observations of temporal and island differences in *P. downsi* behaviour. For example, during 2000 to 2004 on Santa Cruz Island, *P. downsi* were only found in Darwin's finch nests with chicks (100% prevalence), but, since 2012 on Santa Cruz Island, *P. downsi* have regularly been found in Darwin's finch nests with eggs (80% prevalence in some species and some years) and in most nests with chicks (83% to 100% prevalence) [35], indicating adult *P. downsi* are ovipositing earlier in the nesting cycle. By contrast, on Floreana Island, during 2004 to 2016, *P. downsi* larvae and pupae were uncommon in nests with eggs (2% prevalence) but were found in all highland nests with chicks (100% prevalence) [36].

Currently, *P. downsi* is considered the biggest risk factor to the survival of all Galápagos land birds, and researchers are urgently interested in identifying causative factors that explain *P. downsi* oviposition behaviour in host nests, which could be used to inform both mechanical removal and biocontrol approaches [31,37]. Two main hypotheses have been proposed for how *P. downsi* locate host nests: (i) olfactory cues from the host and/or (ii) visual cues from host behaviour [37]. One observation at one Galápagos flycatcher (*Myiarchus magnirostris*) nest provides a clue that parental presence at, or near, a nest entrance may deter an ectoparasite from entering the nest [38]. In this detailed observation, two adult *P. downsi* females were observed resting approximately 50 cm from the nest entrance for approximately 40 min [38]. During a second observation period later in the day, after the attending female flycatcher had left the nest, one of the two *P. downsi* females entered the nest [38]. The flycatcher returned shortly thereafter; both fly and flycatcher remained in the nest for 8 min, then the flycatcher emerged and so did *P. downsi*, flying over the head of the flycatcher [38]. From in-nest video material in Darwin's small ground finch (*Geospiza fuliginosa*), *P. downsi* females were observed to enter and oviposit on the nesting material when the attending female was out of the nest [23]. Fly visitation length at the nest averaged 1.3–1.5 min and was terminated if the adult host returned [23].

We are interested in whether parental care in Darwin's finch hosts could attract or deter *P. downsi* from ovipositing in nests. We do not measure oviposition behaviour directly but use total number of *P. downsi* (larvae, pupae and puparia) per nest as a proxy for total oviposition behaviour. We test the following ideas. (i) If parental activity attracts *P. downsi* via visual cues, then we predict more *P. downsi* in nests with high levels of parental activity (many incubation events, many male and female visits with food delivery to chicks). (ii) If parental attendance at the nest attracts *P. downsi* via other cues, perhaps olfactory cues, then we predict more *P. downsi* in nests with greater parental nest attendance (longer incubation duration of eggs, longer brooding duration of hatchlings, interaction effect between food delivery by male and time inside nest by female). (iii) If parental attendance at the nest deters *P. downsi* from entering the nest, then we predict fewer *P. downsi* in nests with greater parental in-nest attendance (longer incubation duration of eggs, longer brooding duration of hatchlings). As a corollary of this prediction, we explore factors that may be associated with longer female nest attendance (male feeds to female, male feeds to chicks, chick age). Finally, we consider an alternative explanation, namely that the date of nesting explains variation in parental care, given changes in timing of activity associated with ambient temperature [39,40], invertebrate abundance [41] or other factors we did not measure. We also analysed nesting date in relation to number of *P. downsi* per nest, as the number of parasites may increase across the host nesting season [42].

## 2. Methods

### (a) Study site and species
Our study was conducted on two islands in the Galápagos archipelago from January to March during 2000 to 2020. On Santa Cruz Island (−0.624192, −90.384808), we observed Darwin's finch nesting behaviour in 2000, 2001, 2002 and 2004, and on Floreana Island (−1.299829, −90.455674) we observed nesting

behaviour in 2006, 2010, 2012, 2013, 2014, 2016 and 2020 (11 years of sampling spanning 21 years). The two focal species on Santa Cruz Island were small ground finch (*G. fuliginosa*) and small tree finch (*Camarhynchus parvulus*) and the four focal taxonomic groups on Floreana Island were small ground finch, small tree finch, medium tree finch (*C. pauper*) and the recently discovered hybrid *Camarhynchus* group that arises mostly from pairings between *C. pauper* females and *C. parvulus* males [43–45]. Male and female ground finches (*Geospiza* spp.) are easily distinguished once males are 1+ years old, as they become progressively black-bodied with age until attaining a fully black body around 5+ years old whereas females remain olive grey with streaked plumage [46]. In tree finches (*Camarhynchus* spp.), males become progressively black-headed until attaining a fully black crown and hood from around 5+ years old whereas females remain olive green [47,48]. We only consider the effects of male age on parental care because male age can be inferred from plumage colour, whereas female age cannot be inferred from plumage colour. In both ground and tree finches, only females incubate eggs and brood hatchlings [49]. Thus, one can easily discern the sexes of nesting Darwin's finches. Minimum longevity (calculated as the age at first capture plus the number of years until the last recapture) in these finches is 12–17 years [48].

## (b) Parental care behaviour

We analysed 541 1 h nest observations at 208 Darwin's finch nests during the incubation ($n = 305$) or chick feeding ($n = 236$) phase. The number of nests monitored per species was 59 *G. fuliginosa*, 79 *C. parvulus*, 58 *C. pauper* and 12 nests of hybrid *Camarhynchus* finches. We regularly mist-netted birds in the study area during the first two weeks of every field season and then subsequently located nests with colour-banded birds. At 149 nests, at least one bird per pair had unique colour bands, and, at 74 nests, both the male and female were colour-banded. In total, 110 males and 113 females were uniquely colour-banded (26 male and 25 female *G. fuliginosa*; 57 male and 67 female *C. parvulus*; 18 male and 11 female *C. pauper*; 9 male and 12 female hybrid finches).

Nests were monitored using our standardized protocol developed in 2000 and maintained throughout the study [39,43]. To confirm nesting activity, nests were routinely inspected (with binoculars and ladder from 2000 to 2006 and using a borescope since 2008) every 3 days during incubation and every 2 days during the nesting phase. During 1–2 days per phase (incubation, chick feeding), we made a 1 h focal sample observation (between 7:00 and 11:00), noting every behaviour with a time stamp. In about 15% of observations, the observation time was slightly shorter or longer than 60 min. This was accounted for in our analyses by applying the offset function within our statistical models [50]. The observer was seated on the ground, about approximately 20 m from the nest, with binoculars focused on the nest entrance. As Darwin's finches do not noticeably alter their behaviour in the presence of a human observer within 5 m [51], this observer distance was unlikely to disturb parental feeding. During the incubation phase, we recorded the following behaviours: (i) time female spends inside the nest during incubation; (ii) number of incubation events (number of bouts of incubation per hour of observation); (iii) incubation bout duration (min); and (iv) number of male food deliveries to the incubating female at or near the nest entrance. Behaviours recorded during the chick feeding phase (eggs hatched into chicks) were: (v) time female spends inside the nest sitting on hatchlings to provide warmth during the chick phase (brooding duration, min h$^{-1}$); (vi) number of male visits per hour to the nest with food delivery to chicks; and (vii) number of female visits per hour to the nest with food delivery to chicks. We noted clutch size, brood size

and chick age at the time of behavioural observations for all nests from routine nest inspections.

## (c) Number of *Philornis downsi* per nest

Once the nesting event had finished (i.e. the offspring had died or fledged), as confirmed by our routine nest monitoring, the nest was immediately collected from the field, stored in a plastic bag and transported to our on-site laboratory later in the day to count the number of *P. downsi* larvae, pupae and puparia [52]. Generally, all *P. downsi* were counted within 6 h and a maximum of 24 h post-nest collection. The *P. downsi* specimens from each nest were preserved in separate tubes containing 70% ethanol. We recorded the number of *P. downsi* per instar phase, though first and second instar were confirmed later with a microscope [52,53]. Here, we analysed the total number of *P. downsi* per nest as our main variable of interest and in relation to chick age at the time of nesting termination.

## (d) Statistical analysis

All statistical analyses were performed in R v. 3.6.1 [54] under a pseudo-Bayesian framework with non-informative priors using the packages 'arm' [55] and 'lme4' [50]. For every statistical model (package 'lme4'), the restricted maximum-likelihood estimation method was applied, and all the assumptions were checked by visual inspection of the residual plots. In each model, we applied the function 'sim' and carried out 10 000 simulations to obtain the posterior distribution of every estimate, the mean value and the 95% credible interval (CrI) [55]. CrIs provide information about uncertainty around the estimates. We defined a difference between species (and hybrid birds) to be statistically meaningful when the posterior probability of the CrI difference (termed '*p(dif)*') was higher than 95%, and an effect to be statistically meaningful when the 95% CrI did not overlap with zero. A threshold of 5% is equivalent to the significance level in a frequentist framework (i.e. *p*-value of 0.05) [56].

## (i) Effect of incubation and brooding behaviour on number of *Philornis downsi* per nest

To examine the effect of incubation and brooding behaviour on the number of *P. downsi*, we fitted separate Poisson generalized linear mixed effects models. The dependent variable was the number of *P. downsi* in the nest at the time of the nesting outcome. We included observation duration as an offset in the model. To measure the effect of incubation behaviour, the explanatory variables were number of incubation events, incubation bout duration, the interaction between these two variables and number of male food deliveries to the female. To measure the effect of brooding duration, the explanatory variables were female brooding duration percentage, number of female food deliveries to the chicks and number of male food deliveries to the chicks. For better model stability and to obtain standardized effect sizes (i.e. comparable estimates), every dependent variable and covariate was transformed into *z*-scores. We included the interaction terms between (i) brooding duration × male food delivery to chicks, (ii) brooding duration × female food delivery to chicks and (iii) male food delivery × female food delivery to the chicks. Year, time of observation and nest ID nested within species were included as random factors to account for among-year variation, short-term environmental variation, among-species variation, and repeated measures, respectively. We are aware that parental age and breeding experience could influence brooding and incubation behaviours [57], though this is not always the case [58]. However, our sample size did not allow for including additional covariates in our models. To account for this, we carried out a separate analysis quantifying the effect of male age on the different parental care behaviours

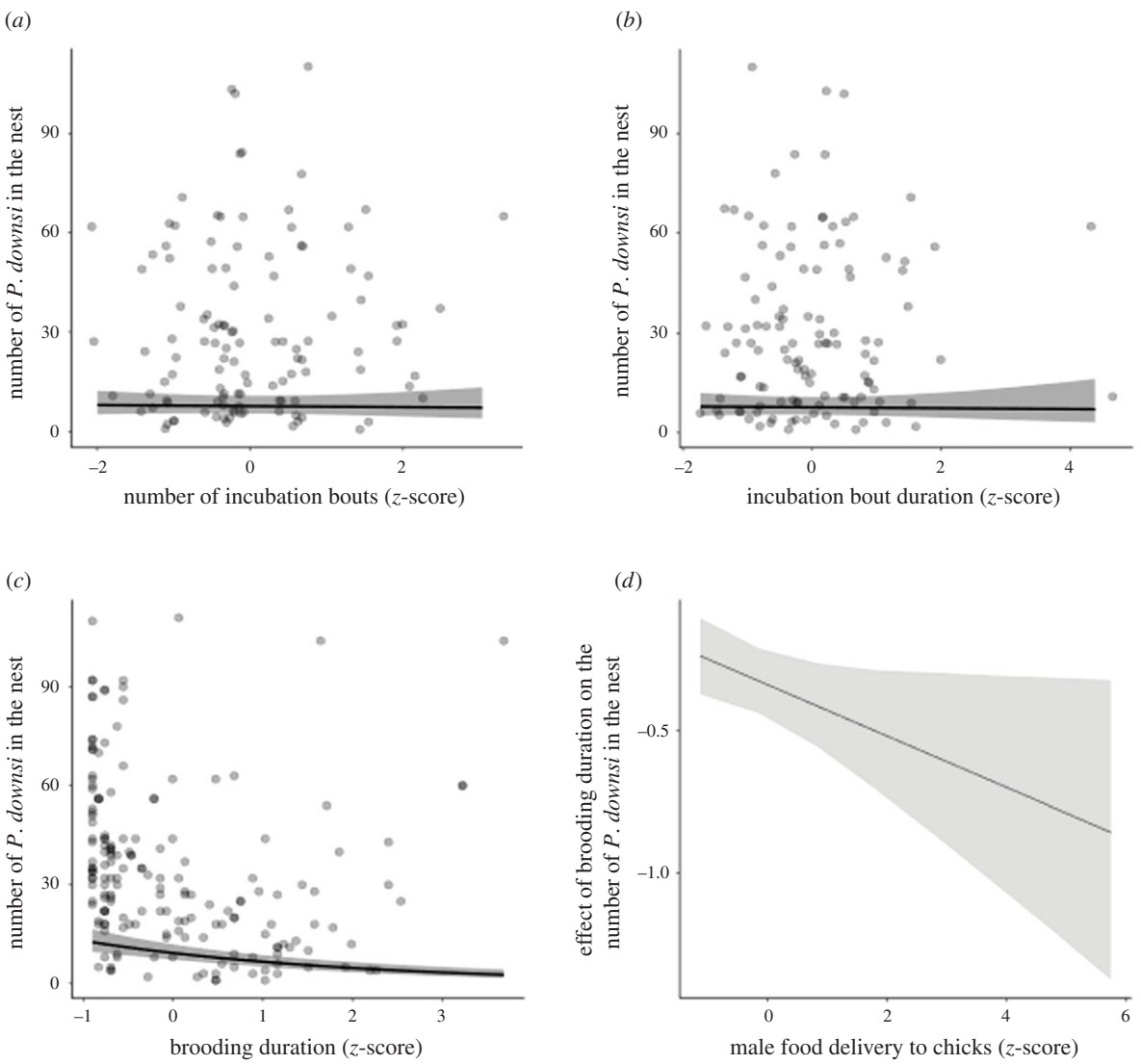

**Figure 1.** Female incubation and brooding behaviour, number of *Philornis downsi* in Darwin's finch nests and male food delivery to chicks. (*a*) Relationship between number of female incubation bouts (per hour) and number of *P. downsi* in the nest. (*b*) Relationship between female incubation bout duration (min) and number of *P. downsi* in the nest. (*c*) Relationship between female brooding duration (min) and number of *P. downsi* in the nest. Longer female brooding duration is associated with fewer *P. downsi* per nest but depends on male provisioning behaviour. (*d*) Interaction between number of male food deliveries to the chicks and female brooding duration. Males that provided more food deliveries to the chicks enhanced the negative association between female brooding duration and number of *P. downsi* in the nest. Black lines represent the mean estimate, grey ribbons the 95% CrIs and grey dots the raw data.

across species and found that this effect was, if any, very small (see electronic supplementary material, figure S1 and table S1 for details). Further, for a descriptive purpose and presentation of the raw data, variation in the parental care parameters for each species were quantified in separate models (see electronic supplementary material, figure S2 and table S2 for details).

To explore possible differences across islands, we ran the same models with the addition of island as covariate. For the model of brooding duration, species was removed as a random factor because it explained very little variance, and with it the complete model including island would not converge. The reason for running these models separately was because the addition of island as a covariate pushed the number of explanatory variables, covariates and interactions to the maximum limit allowed by our sample size.

In a separate model, we explored the effects of chick age and brooding duration on the number of *P. downsi*. Here, we carried out a Poisson generalized linear mixed effects model with number of *P. downsi* as the dependent variable. The explanatory variables were brooding duration percentage, chick age, the interaction between brooding duration × chick age and island. As with the previous models, (i) numerical dependent variables

and covariates were converted to z-scores, and (ii) year, time of observation and nest ID nested within species were included as random factors.

## 3. Results

### (a) Parental care and *Philornis downsi* parasitism

Within the incubation phase, there was no evidence that the number of female incubation events (figure 1*a* and table 1), number of male food deliveries to the incubating female (table 1), or female incubation bout duration (figure 1*b* and table 1) affected the number of *P. downsi* in the nest (electronic supplementary material, table S3). These results during the incubation phase were consistent across both islands (electronic supplementary material, table S4).

During the chick feeding phase, female food delivery to chicks was not associated with the number of *P. downsi* in the nest (table 2). Brooding duration had a negative effect on the number of *P. downsi* when all covariates are at their

**Table 1.** Effect of incubation behaviours on *P. downsi* parasitism. The response variable number of *P. downsi* was modelled with a Poisson error distribution. Estimates of fixed ($\beta$) and random ($\sigma^2$) effects with their 95% CrIs are shown in brackets. Values of '0.00' represent values smaller than 0.001. We found no statistically meaningful effects (i.e. all the CrI overlap zero) of any incubation behaviour on the number of parasites in the nest.

| fixed effects | no. of *P. downsi* $\beta$ (95% CrI) |
| --- | --- |
| intercept | 2.03 (1.78; 2.38) |
| no. of incubation events | −0.02 (−0.17; 0.13) |
| incubation bout duration | −0.02 (−0.18; 0.14) |
| no. of male food delivery to female | −0.06 (−0.14; 0.02) |
| no. of incubation events × incubation bout duration | −0.001 (−0.07; 0.06) |
| **random effect** | **$\sigma^2$ (95%CrI)** |
| year | 0.14 (0.05; 0.32) |
| species | 0.006 (0.001; 0.02) |
| nest ID | 1.1 (0.83; 1.43) |
| time of observation | 0.008 (0.005; 0.01) |

mean (figure 1*a* and table 2), but this effect depended on male provisioning to chicks (interaction term in table 2). Specifically, when males delivered more food, the effect of brooding duration on number of *P. downsi* was stronger (figure 1*b*). Also, this effect was attenuated with increasing age of the chicks and disappeared after the chicks were older than 6 days (figure 2; interaction term, electronic supplementary material, table S4), a period in which females brooded less (electronic supplementary material, figure S3). We found the same relationship between brooding duration and *P. downsi* on both islands (electronic supplementary material, table S5, and figure S4).

The effect of brooding on the number of *P. downsi* remained statistically meaningful while accounting for the date of nesting onset (electronic supplementary material, table S6) and the date of nesting onset was not associated with female brooding duration and/or the number of parasites in the nest (electronic supplementary material, table S7). Finally, we found no evidence for shifts in parental behaviour (i.e. female brooding duration, male food deliveries, female food deliveries) across the years (electronic supplementary material, table S8).

## 4. Discussion

The temporal window for ectoparasite oviposition behaviour can be influenced by parental care, which may alter the course of age-specific costs to offspring survival from ectoparasites. If temporal patterns of host parental care differ across the sexes—for example, in systems with uniparental male or female in-nest attendance—the threat of parasite oviposition may increase sex-specific costs of parental care to the attending parent [59,60]. In this study on Darwin's finches parasitized by *P. downsi* (Diptera: Muscidae), longer female brooding duration during the first days post-hatch was associated with fewer *P. downsi.* Increased male food delivery

to chicks strengthened the effect of longer brooding duration on fewer *P. downsi*, irrespective of male age or species. There was no effect of nesting date on patterns of parental care or number of *P. downsi*, and no effect of parental care during the incubation phase on number of *P. downsi*.

We found no effect of year on parental care variables in this study, which is perhaps surprising given strong natural selection by *P. downsi*. There are several possible reasons why female brooding duration did not lengthen over time due to directional selection. Opposing environmental factors such as resource quality and abundance, thermal risk and predation risk, which we did not measure, can potentially impact parental care decisions in a given year [3]. Further, there is no evidence in birds that brooding duration or food delivery are heritable traits [61,62]. Finally, given that Darwin's finches can live to approximately 17 years, our generational sampling time window may be too shallow to detect such an evolutionary change should it occur, as the parents survive *P. downsi* but the offspring die [63].

Understanding mechanisms related to the timing of parental care and temporal windows of costs and benefits from parental care that increase or reduce parasite burden will shed light on evolutionary pressures that favour parental care traits. For example, in non-social *Ammophila* wasps, larvae may be parasitized by cuckoo flies (Diptera: Miltogramminae) that gain access to the wasp larvae when females open the nest to tend the larvae [64]. Female wasps that progressively provision offspring and abandon parasitized offspring lose less investment than females that mass provision each larva at oviposition [64]. The *Ammophila* wasp study provides experimental evidence for an adaptive value of progressive provisioning across the nesting cycle, as well as benefits of nest guarding to avoid costs of parasitism in a uniparental carer. Our study provides evidence in some songbirds that coordinated female and male parental care may be favoured when longer in-nest attendance by females and increased food provisioning by males deters ectoparasites from entering the nest.

We acknowledge that longer in-nest attendance could have many functions, including thermal insulation of chicks [65], defence against nest predators [66] and removal of ectoparasites via nest sanitation [23,24]. We consider female nest sanitation an unlikely explanation for our consistent finding of fewer *P. downsi* in nests with longer female brooding duration, for two reasons. Firstly, the 1st instar *P. downsi* larvae reside inside the nares of chicks and emerge to feed externally on the chicks from d2 to d4 onwards [53]. We may expect an increase in brooding (and nest sanitation) later in the nesting cycle (i) after *P. downsi* have emerged from the chicks' nares around d3 and (ii) when there are more *P. downsi* in the nest from d3 onwards (electronic supplementary material, figure S3). However, rather than finding an increase in female brooding duration after d3 as the number of *P. downsi* increased, we found a stronger impact of early female brooding duration on fewer *P. downsi* (figure 2), and as observed generally in songbirds, a decrease in female brooding duration with chick age (electronic supplementary material, figure S3). We suggest that female presence at the nest early post-hatch deters *P. downsi* from entering the nest to oviposit. Further, we predict that the fitness benefit should be maximized when females prevent parasites from ovipositing on the youngest chicks, which are small, defenceless, and still unable to self-preen. Our statistical analysis,

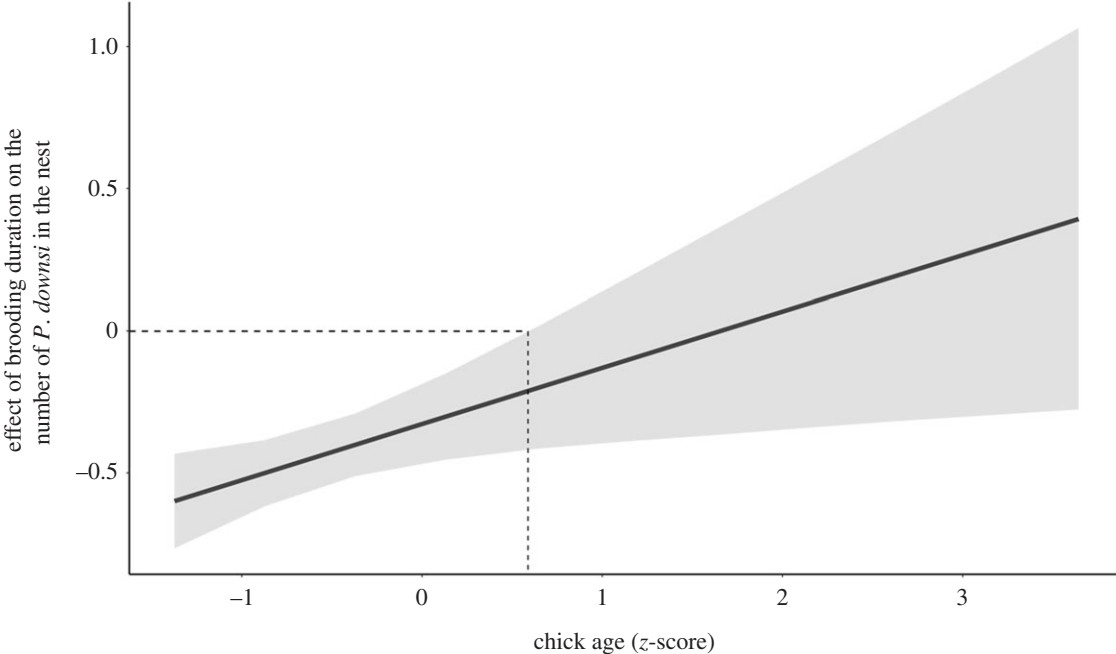

**Figure 2.** Interaction between chick age (*z*-score) and brooding. The effect of brooding on the number of *Philornis downsi* was attenuated with increasing age. When the 95% CrI overlaps zero, there is no effect of brooding duration on the number of *P. downsi*, and this is the case for chicks older than 6 days (dashed line). Black lines represent the mean estimate and grey ribbons the 95% CrIs. Details on this model's estimates can be found in the electronic supplementary material, table S4.

**Table 2.** Effect of brooding behaviours on *P. downsi* parasitism. The response variable number of *P. downsi* was modelled with a Poisson error distribution. Estimates of fixed ($\beta$) and random ($\sigma^2$) parameters with their 95% CrIs are shown in brackets. Statistically meaningful effects are those where the CrI do not overlap cero (i.e. posterior *p* greater than 95%) and are marked in italics.

| fixed effects | no. of *P. downsi* $\beta$ (95% CrI) |
|---|---|
| intercept | 2.27 (1.98; 2.48) |
| brooding duration (%) | *−0.33 (−0.45; −0.22)* |
| no. of female food delivery to chicks | 0.11 (−0.004; 0.21) |
| no. of male food delivery to chicks | −0.03 (−0.12; 0.06) |
| brooding duration (%) × no. of female food delivery | 0.01 (−0.10; 0.08) |
| brooding duration (%) × no. of male food delivery | *−0.09 (−0.17; -0.01)* |
| no. of male food delivery × no. of female food delivery | 0.003 (−0.08; 0.07) |
| **random effect** | $\sigma^2$ **(95%CrI)** |
| year | 0.06 (0.03; 0.11) |
| species | 0.00 (0.00; 0.00)[a] |
| nest ID | 0.69 (0.57; 0.83) |
| time of observation | 0.09 (0.06; 0.11) |

[a]0.00 indicates value smaller than 0.0001.

which controlled for the potential confounding effect of chick age on brooding duration and number of *P. downsi*, supports this view.

If in-nest attendance buffers against ectoparasites entering the nest to oviposit, then we predict stronger selection on coordination of male and female behaviour to enhance nest guarding. The mechanism by which choosy females could predict within-pair coordination is unknown, but future research could explore signals of male quality, such as rhythmicity of song or other pre-mating interactive behaviour between the male and female [67–71]. Across systems, females have been shown to increase parental care when paired with a high-quality male [72,73], whereby older males may signal quality because of experience and capacity

to survive [74]. The minimum longevity of 12–17 years in Darwin's finches [48] is rather long for small songbirds (13–18 g). Females paired with older males could receive direct benefits, including a safer nest site, increased male vigilance and/or increased food delivery to the female [47,75,76]. Females paired with older males may also receive indirect benefits, including good genes for their offspring or increased male provisioning of offspring [74,77]. In this study, there was a mix of results for direct and indirect benefits to females paired with older males. For example: (i) in hybrid nests, older males provided more food deliveries to the incubating female (direct benefit) and females paired with older males had longer brooding duration (indirect benefit; see electronic supplementary material, figure S2); (ii) in *C. parvulus* nests, females paired with older males had longer incubation duration (indirect benefit, see electronic supplementary material, figure S2) and (iii) in *G. fuliginosa* nests, older males provided more food deliveries to chicks (indirect benefit; see electronic supplementary material, figure S2).

Reduced ectoparasite burden from longer female in-nest attendance may create evolutionary incentives for males to provide more food deliveries to attending females, or for increased male incubation, brooding and/or helper care. For example, male Moustached Warblers (*Acrocephalus melanopogon*) contribute to incubation even though they do not possess fully developed brood patches; they prevent egg cooling and provide a form of nest guarding [66,78], and helper males provide extra incubation and supplemental feeds to chicks [79,80]. Helper behaviour was also observed at Darwin's finch nests during 1979 on Daphne Major Island. At 11 of 21 Cactus Finch (*G. scandens*) nests, chicks were fed 1–10 times per day by a visiting conspecific male; these helper males provided between 1.7% and 24.9% of all food regurgitations at a nest per day, had comparable feeding rates to the paternal male and, in four cases, also removed faecal sacs from the nest [81]. Thus, there exists behavioural plasticity in parental feeding care in some Darwin's finch groups that could be the target of selection should helper behaviour become advantageous.

Cooperative breeding in birds is widespread in systems that experience high levels of brood parasitism [82]. Helper birds in cooperatively breeding groups may provide additional nest guarding that deters brood parasites from entering the nest to oviposit [83], a form of frontline defence [84–86]. Future research could explore proximate and ultimate explanations for nest guarding against brood parasites and ectoparasites [87]. Since female *P. downsi* flies probably use visual or olfactory cues from the nest to locate and infect avian hosts, nest guarding has the potential to be an effective strategy to prevent infection by *P. downsi*, but experimental studies are required to unveil this causal relationship.

Although both islands showed the same relationship between female brooding duration and number of *P. downsi*, the Santa Cruz data were collected from 2000 to 2004 and the Floreana Island data were collected from 2004 to 2020. Since 2004, there have been measurable shifts in *P. downsi* and host behaviour on both islands. *Philornis downsi* oviposition and hatching have shifted towards earlier in the host incubation period on Santa Cruz Island, whereas *P. downsi* hatching (and perhaps oviposition) occurs during chick feeding on Floreana [35,52,88]. On Floreana Island, the

proportion of hybrid tree finches in the population increased to 55% in 2006 and has since remained at approximately 40% [43,45], and higher admixture (e.g. greater introgression among *C. parvulus* and *C. pauper*) was associated with fewer *P. downsi* per nest [44,89] and different patterns of parental care (this study). All *P. downsi* life stages measured on Santa Cruz and Floreana Island between the years 2004 and 2020 have become approximately 20–30% smaller [52], pointing to coevolutionary dynamics. This current study identifies effects of parental care that may be associated with changes in fly oviposition behaviour and success, which could produce new host and parasite equilibria across islands and time [90].

In conclusion, the results of this study show that the number of *P. downsi* in nests was not related to the number of host visits, but was negatively related to female in-nest attendance during the brooding period. This case study presents evidence that an introduced ectoparasite may select for longer in-nest attendance by host females to physically prevent parasites from ovipositing (with potentially higher costs to host females than males) and leads to the prediction that there may be selection for coordinated host male and female behaviour to guard the nest when presence at the nest safeguards the nesting contents. Using modelling approaches, studies have found evidence for cross-scale feedbacks between host resource abundance and parasite transmission [91] and shifts from fluctuating to directional selection dynamics that drive greater genetic divergence between host populations [92]. Our study highlights the role of host parental care behaviour as one factor that may affect the timing of parasite oviposition behaviour and may therefore play a role in shaping selection dynamics in an emerging host–parasite system.

Ethics. The work was approved by the Flinders University Animal Welfare Committee (E270, E393, E480-19).

Data accessibility. Data are available from the Dryad Digital Repository: https://doi.org/10.5061/dryad.mw6m905x6 [93].

Authors' contributions. S.K.: conceptualization, data curation, formal analysis, funding acquisition, investigation, methodology, project administration, resources, supervision, validation, visualization, writing—original draft, writing—review and editing; L.K.C.: data curation, investigation, writing—review and editing; J.A.O.: data curation, investigation, writing—review and editing; J.G.-L.: investigation, writing—review and editing; A.C.K.: investigation, writing—review and editing; R.Y.D.: investigation, supervision, writing—review and editing; D.C.-N.: data curation, investigation, project administration, supervision, writing—review and editing; N.M.A.: formal analysis, methodology, software, validation, writing—review and editing. All authors gave final approval for publication and agreed to be held accountable for—work performed therein.

Competing interests. We declare we have no competing interests.

Funding. This work was supported by Australian Research Council (grant no. DP190102894), Rufford Small Grant Foundation, Mohamed bin Zayed Species Conservation Fund, Max Planck Institute for Ornithology, Earthwatch Institute, Galápagos Conservation Fund and Macquarie University.

Acknowledgements. Permission to conduct this study was granted by the Galápagos National Park Directorate (PC-021-99, PC-19-07, PC-39-09, PC-58-11, PC-38-12, PC-15-14, PC-23-16, PC-02-20) with logistical support provided by the Charles Darwin Research Station. We thank all volunteers and students who have contributed to the project over time, as well as the Floreana Island community for support. This publication is contribution number 2369 of the Charles Darwin Foundation for the Galápagos Islands.

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
