## [Peer Review File · Proceedings of the Royal Society B: Biological Sciences]

Review History

RSPB-2021-1668.R0 (Original submission)

Review form: Reviewer 1 (Heinz Richner)

Recommendation

Accept with minor revision (please list in comments)

Scientific importance: Is the manuscript an original and important contribution to its field?

Excellent

General interest: Is the paper of sufficient general interest?

Excellent

Quality of the paper: Is the overall quality of the paper suitable?

Excellent

Is the length of the paper justified?

Yes

Should the paper be seen by a specialist statistical reviewer?

No

Do you have any concerns about statistical analyses in this paper? If so, please specify them explicitly in your report.

No

It is a condition of publication that authors make their supporting data, code and materials available - either as supplementary material or hosted in an external repository. Please rate, if applicable, the supporting data on the following criteria.

Is it accessible?

Yes

Is it clear?

Yes

Is it adequate?

Yes

Do you have any ethical concerns with this paper?

No

Comments to the Author

Overall, this is a very good study in terms of long-term data, content, analysis, and writing. The study is relevant regarding the urgent conservation needs for saving the unique Darwin finches in the Galapagos.

The main finding of the study is that broods with longer female brooding duration have fewer blood sucking parasitic fly larvae, and that the effect increases when males deliver more food (i.e. a main effect and a interaction effect being significant). The main proposed hypothesis for this finding is that female nest attendance deters parasitic flies from ovipositing in a host nest. While this is a non-experimental study where cause and consequence by definition remain unclear, there is some observational support for this hypothesis. Also the study is non-experimental, it is still important as a first demonstration of such a relationship between nest attendance and parasite numbers, and will clearly trigger many subsequent studies into the mechanisms and selection pressures responsible for the relationship.

In my view, the main alternative hypothesis that should and can be evaluated with the available data is the following: In many bird species with resource-dependent territorial defence, the early breeding birds within the annual breeding season are the high-quality birds, plus early in the breeding season there are also much fewer parasites seeking hosts. Thus, the main finding here could be simply due to the co-variable given by the timing of breeding. I think this needs to be included into the analysis, also for guiding subsequent studies.

In addition to parasite pressure, the timing of bird breeding evolved to occur at times of highest abundance and quality of food. Hence, early in the annual breeding season, a female can attend nests for longer without a change in male food provisioning rates.

The abstract (and paper) needs some clarifications:

Food delivery: There is nowhere a description how food delivery has been measured, for example how, when, how long and how often. Furthermore, it is not stated what the variable expresses, hourly rate, daily rate, total rate over the brooding period, or total number of food items over the entire brooding period? The latter would be strongly influenced by the main significant variable in the study, i.e. brooding duration.

Brooding duration: The reader cannot figure out from the abstract, what the main significant variable exactly describes. In the methods only it is stated as "time female spends inside the nest

during the chick phase (brooding duration, min per hr)". This needs to be made clear in the abstract. In a following sentence of the abstract it says "female nest attendance duration" but it is not made clear whether this is the same as brooding duration.

Prediction for selection for coordinated male and female nest attendance: In the Methods (line 154) it is stated that only females incubate eggs and hatchlings. It then remains unclear whether this prediction refers to just the hatchling period or to the entire nestling phase. It is further confusing because "brooding duration" refers to the entire chick (i.e. nestling) phase (see above).

The Discussion of this paper has a strong evolutionary tack that Darwin would have liked.

Review form: Reviewer 2

Recommendation

Major revision is needed (please make suggestions in comments)

Scientific importance: Is the manuscript an original and important contribution to its field?

Acceptable

General interest: Is the paper of sufficient general interest?

Acceptable

Quality of the paper: Is the overall quality of the paper suitable?

Acceptable

Is the length of the paper justified?

Yes

Should the paper be seen by a specialist statistical reviewer?

No

Do you have any concerns about statistical analyses in this paper? If so, please specify them explicitly in your report.

Yes

It is a condition of publication that authors make their supporting data, code and materials available - either as supplementary material or hosted in an external repository. Please rate, if applicable, the supporting data on the following criteria.

Is it accessible?

Yes

Is it clear?

Yes

Is it adequate?

Yes

Do you have any ethical concerns with this paper?

No

Comments to the Author

Review for: Female in-nest attendance reduces number of ectoparasites in Darwin's finch species
 Proceedings of the Royal Society B
 August 2021

Overview: This manuscript examines the relationship between parental care and the abundance of an introduced nest parasite, the avian vampire fly, in three species of Galapagos finches. The study looked at several types of parental care across the nestling developmental period for both male and female parents: including incubation, brooding, and provisioning. The dataset of observations from 208 nests spans a 21-year period (11 years with data). The study finds that nests with longer female brooding durations during the early phase of nestling development (under 6 days old) had fewer flies. This effect was intensified in nests where there were more male food deliveries to nestlings. There were no significant relationships between other measures of parental care and the number of fly larva. The authors suggest that female brooding behavior during this key period likely limits the ability of adult flies to oviposit into the nests, thus limiting parasite abundance and that brooding and potentially coordination of male and female care in this case can be seen as a form of nest guarding.

I appreciate the large sample size and many years of data presented in this study, which represent an impressive amount of field work, as well as the detailed look at parental care, which included different types of care for both males and females. This study is also novel as it examines the role of nest attendance as a parasite deterrent, while previous work has largely focused on the role of provisioning and food compensation with nest parasites (as the authors point out). I offer suggestions to improve the study as well as point out areas that need clarification below.

Major Comments:

-I think more natural history information about the fly needs to come earlier in the paper. For readers who are unfamiliar with this system, that information is key to understand predictions for what types of parental care may be important to deter ovipositing and when during development that occurs. The authors should discuss when and how flies are targeting nests (what is known), the duration that flies target nests during the nestling developmental period. The authors do discuss this very briefly in the discussion (lines 358-361)- and for me, this made it much clearer why they were looking at incubation in addition to brooding and provisioning (initially, I assumed flies would only target nests with nestlings so was unsure why they were looking at parental care during the egg phase). The fact that there have been measurable shifts through time in when flies are targeting nests and presumably the cost of flies (flies are getting smaller) is also relevant to this study and should be brought up earlier in the paper so readers can understand the results in context, especially because this dataset spans both islands and an extended period of time (21 years).

-Building off the previous comment- I think it is interesting that there have been measurable shifts in fly behavior (timing of oviposition) and cost through time and wonder if the study could further explore that with regards to parental behavior. If I am understanding the results correctly, the random effect of year, particularly for the brooding model presented in the main text is significant. I would be curious if parental care is shifting through time, and if this nest guarding aspect of nest attendance is becoming stronger as the birds adapt to this parasite? This is particularly interesting given the context that this is an introduced parasite.

-I had several questions about the statistics, some of which the authors may have done, and it was just unclear, and others that may suggest changes to the models.

- The initial statistical analysis described in the methods says (lines 204-205) "A gaussian error distribution was assumed for every model..." yet the models described in the main text use count data and are stated as being Poisson models? Make it clearer which models are presented in the main text and which are presented in the supplement.
- There are times when certain numerical fixed effects in a model were z-transformed

while others were not (for example, (lines 223-224 “To measure the effect of brooding duration, the explanatory variables were female brooding duration % (transformed to z-scores), number of female food deliveries to the chicks, and number of male food deliveries to the chicks.”). To improve the stability of the models and to be able to directly compare the beta estimates of different fixed effects within the same model all numerical fixed effects should be z-transformed. This will put all the different measures on the same scale so effects can be more accurately estimated.

- The random effects of the models in the main text include year, species, nest ID, and time of observation. Is there a reason the authors did not nest “Nest ID” within “species”?
- For the results presented in the main text as well as in the supplement, the models include interaction terms. Typically, the main effects cannot be clearly interpreted while those same terms are included in an interaction term in the model, but the authors appear to do this. I do not think these rules would change for their quasi-Bayesian framework, but I could be mistaken. Were non-significant interactions removed, and the models rerun? How were main effects interpreted when interactions were significant? For example, how are the authors interpreting a “significant” main effects (such as brooding duration) when that term is also included in two different interaction terms in the same model, one of which is significant (table 2)?
- In the supplement, the authors describe several additional models that were run. This includes (lines 423-427) “... we fitted one linear mixed effect model for each of the six dependent behaviors (incubation duration %, # incubation events, male food deliveries to the female, brooding duration, male food deliveries to the chicks, female food deliveries to the chicks).” Yet a linear mixed model is not appropriate for all these different response variables. For example, % time would be better modeled with a general linear mixed model with a beta distribution. If the number of incubation events per hour had to be square root transformed to use a linear model, would a different distribution be a better fit for the raw data?
- According to the methods, the abundance of flies was measured once when nests either failed or nestlings fledged (lines 191-200). Looking at figure S3, it becomes clear that these measurements were taken across the nestling period (nests failed at 2-11 days, or fledged? As there are fly measurements during all these points). Was whether a nest failed or fledged, and the timing of failure considered in the statistical analysis? Presumably nests that failed early would have less time to accumulate flies compared to nests that failed later or fledged? If nests were heavily parasitized, females may have abandoned nests, or reduced nest attendance and then abandoned- could this be driving the behavior patterns that you are seeing?

-I agree with the authors that parental age can have important impacts on behavior and that it should be examined if the data are available. Currently, the authors examine male age, but it is not mentioned why they do not examine female age (unless I missed something)? Because female behavior had the largest effect on parasites abundance, it seems that female age would possibly be more important in this case. Were the data unavailable?

-In the discussion, the authors predict strong selection on coordination of male and female behavior to enhance nest guarding. While they suggest that this is an area for future study, could they not test this with their data? If I understand things correctly, they looked at male and female behavior (male feeding, female brooding) and the interaction- but could they not take their data and look at the proportion of time that the nest was left unattended to look at coordinated nest guarding by both parents? For example, if males were visiting the nest to provision offspring when the female was absent?

Minor Comments:

-The authors do point out in the introduction that much work has been done on the role of parental provisioning and food compensation with avian nest parasites, and that the role of nest-attendance as a form of nest guarding against parasites is a fairly novel area. But I found the sentence in the abstract: (Lines 42-42) “Such predictions have been robustly tested in predation risk contexts, but little is known about parental care investment trade-offs under conditions of

parasitism.” to be inaccurate. Nest parasite studies and manipulations have been done for decades and have become a standard model for testing the tradeoffs of parental care, parent-offspring conflict, and eco-immunology.

-I am curious if there is evidence for nest guarding or the role of nest attendance with preventing egg laying by avian brood parasites. If so, this could be a useful parallel to draw in the discussion. The authors do bring this up briefly with regards to cooperative breeding, but I think there may be more there.

-While the authors make a fairly good case that female nest-attendance is likely reducing egg laying by flies and thus reducing the abundance of parasites in the nest, this study is correlational, and this hypothesis remains to be thoroughly tested. It could be that females are just investing less in heavily parasitized nests. Given this, I would suggest changing the title slightly to more appropriately fit their study.

-The authors often refer to “male food delivery” without clarifying whether this was to females or to nestlings. Since they measured both, I think it is important to clarify throughout to avoid confusion.

-At times, particularly in the discussion, the writing can be a bit hard to follow. I think ideas could be better connected and flow more clearly within paragraphs.

-I am not familiar with what minimum longevity mean? Is this not just average adult lifespan?

-Lines 262-263. It seems that some numbers may have been flipped here or I am confused. “But females with longer brooding duration had, on average, 3.8 x fewer *P. downsi* (mean number of *P. downsi* [95% CrI] = 11.8 [7.3; 19.2]) than females with shorter brooding duration (mean number of *P. downsi* [95% CrI] = 3.1 [1.8; 5.2] | Fig. 1A, Table 2)” 11.8 is bigger than 3.1?

-Lines 328: missing “to”

-Is it possible to include raw data points in the main text figures? This would be more transparent and informative.

Decision letter (RSPB-2021-1668.R0)

23-Aug-2021

Dear Dr Kleindorfer:

Your manuscript has now been peer reviewed and the reviews have been assessed by an Associate Editor. The reviewers’ comments (not including confidential comments to the Editor) and the comments from the Associate Editor are included at the end of this email for your reference. As you will see, the reviewers and the Editors have raised some concerns with your manuscript and we would like to invite you to revise your manuscript to address them.

Research ethics:

Use of animals and field studies:

It is a condition of publication that you make available the data and research materials supporting the results in the article. Please see our Data Sharing Policies (<https://royalsociety.org/journals/authors/author-guidelines/#data>). Datasets should be deposited in an appropriate publicly available repository and details of the associated accession number, link or DOI to the datasets must be included in the Data Accessibility section of the article (<https://royalsociety.org/journals/ethics-policies/data-sharing-mining/>). Reference(s) to datasets should also be included in the reference list of the article with DOIs (where available).

All supplementary materials accompanying an accepted article will be treated as in their final form. They will be published alongside the paper on the journal website and posted on the online

figshare repository. Files on figshare will be made available approximately one week before the accompanying article so that the supplementary material can be attributed a unique DOI. Please try to submit all supplementary material as a single file.

Please submit a copy of your revised paper within three weeks. If we do not hear from you within this time your manuscript will be rejected. If you are unable to meet this deadline please let us know as soon as possible, as we may be able to grant a short extension.

Best wishes,
Dr Sasha Dall
mailto:proceedingsb@royalsociety.org

Associate Editor
Board Member: 1

Comments to Author:

I have now received two reviews of your paper from reviewers who clearly read it carefully, both of whom provided useful feedback. Based on these reviews and my own assessment, I feel that the paper could be suitable for publication in PRSB, providing you can address the reviewers' comments. In particular, I would pay attention to the following:

Reviewer 1's comment that timing of breeding might in fact drive the observed correlation is worth considering – it is a logical alternative hypothesis and seems relatively easy to explore with the data at hand.

Reviewer 2's about the statistics are also important, particularly a) whether gaussian errors are appropriate for count data (usually not – alternatives include Poisson and negative binomial models), and b) being careful about interpreting main effects that are involved in significant interactions, and being clear about how interactions are treated (e.g. are non-significant interactions retained in or dropped from models)

Note that a decision of 'revise' is not an acceptance, it just means that the paper does not necessarily have to (but still could) be sent back to reviewers. PRSB does not allow multiple rounds of editing so I encourage authors to address the comments as thoroughly as they can. I look forward to reading your resubmission.

Reviewer(s)' Comments to Author:

Referee: 1

Comments to the Author(s)

Overall, this is a very good study in terms of long-term data, content, analysis, and writing. The study is relevant regarding the urgent conservation needs for saving the unique Darwin finches in the Galapagos.

The main finding of the study is that broods with longer female brooding duration have fewer blood sucking parasitic fly larvae, and that the effect increases when males deliver more food (i.e. a main effect and a interaction effect being significant). The main proposed hypothesis for this finding is that female nest attendance deters parasitic flies from ovipositing in a host nest. While this is a non-experimental study where cause and consequence by definition remain unclear, there is some observational support for this hypothesis. Also the study is non-experimental, it is still important as a first demonstration of such a relationship between nest attendance and

parasite numbers, and will clearly trigger many subsequent studies into the mechanisms and selection pressures responsible for the relationship.

In my view, the main alternative hypothesis that should and can be evaluated with the available data is the following: In many bird species with resource-dependent territorial defence, the early breeding birds within the annual breeding season are the high-quality birds, plus early in the breeding season there are also much fewer parasites seeking hosts. Thus, the main finding here could be simply due to the co-variable given by the timing of breeding. I think this needs to be included into the analysis, also for guiding subsequent studies.

In addition to parasite pressure, the timing of bird breeding evolved to occur at times of highest abundance and quality of food. Hence, early in the annual breeding season, a female can attend nests for longer without a change in male food provisioning rates.

The abstract (and paper) needs some clarifications:

Food delivery: There is nowhere a description how food delivery has been measured, for example how, when, how long and how often. Furthermore, it is not stated what the variable expresses, hourly rate, daily rate, total rate over the brooding period, or total number of food items over the entire brooding period? The latter would be strongly influenced by the main significant variable in the study, i.e. brooding duration.

Brooding duration: The reader cannot figure out from the abstract, what the main significant variable exactly describes. In the methods only it is stated as “time female spends inside the nest during the chick phase (brooding duration, min per hr)”. This needs to be made clear in the abstract. In a following sentence of the abstract it says “female nest attendance duration” but it is not made clear whether this is the same as brooding duration.

Prediction for selection for coordinated male and female nest attendance: In the Methods (line 154) it is stated that only females incubate eggs and hatchlings. It then remains unclear whether this prediction refers to just the hatchling period or to the entire nestling phase. It is further confusing because “brooding duration” refers to the entire chick (i.e. nestling) phase (see above).

The Discussion of this paper has a strong evolutionary tack that Darwin would have liked.

Referee: 2

Comments to the Author(s)

Review for: Female in-nest attendance reduces number of ectoparasites in Darwin's finch species
 Proceedings of the Royal Society B
 August 2021

Overview: This manuscript examines the relationship between parental care and the abundance of an introduced nest parasite, the avian vampire fly, in three species of Galapagos finches. The study looked at several types of parental care across the nestling developmental period for both male and female parents: including incubation, brooding, and provisioning. The dataset of observations from 208 nests spans a 21-year period (11 years with data). The study finds that nests with longer female brooding durations during the early phase of nestling development (under 6 days old) had fewer flies. This effect was intensified in nests where there were more male food deliveries to nestlings. There were no significant relationships between other measures of parental care and the number of fly larva. The authors suggest that female brooding behavior during this key period likely limits the ability of adult flies to oviposit into the nests, thus limiting parasite abundance and that brooding and potentially coordination of male and female care in this case can be seen as a form of nest guarding.

I appreciate the large sample size and many years of data presented in this study, which represent an impressive amount of field work, as well as the detailed look at parental care, which included different types of care for both males and females. This study is also novel as it

examines the role of nest attendance as a parasite deterrent, while previous work has largely focused on the role of provisioning and food compensation with nest parasites (as the authors point out). I offer suggestions to improve the study as well as point out areas that need clarification below.

Major Comments:

-I think more natural history information about the fly needs to come earlier in the paper. For readers who are unfamiliar with this system, that information is key to understand predictions for what types of parental care may be important to deter ovipositing and when during development that occurs. The authors should discuss when and how flies are targeting nests (what is known), the duration that flies target nests during the nestling developmental period. The authors do discuss this very briefly in the discussion (lines 358-361)- and for me, this made it much clearer why they were looking at incubation in addition to brooding and provisioning (initially, I assumed flies would only target nests with nestlings so was unsure why they were looking at parental care during the egg phase). The fact that there have been measurable shifts through time in when flies are targeting nests and presumably the cost of flies (flies are getting smaller) is also relevant to this study and should be brought up earlier in the paper so readers can understand the results in context, especially because this dataset spans both islands and an extended period of time (21 years).

-Building off the previous comment- I think it is interesting that there have been measurable shifts in fly behavior (timing of oviposition) and cost through time and wonder if the study could further explore that with regards to parental behavior. If I am understanding the results correctly, the random effect of year, particularly for the brooding model presented in the main text is significant. I would be curious if parental care is shifting through time, and if this nest guarding aspect of nest attendance is becoming stronger as the birds adapt to this parasite? This particularly interesting given the context that this is an introduced parasite.

-I had several questions about the statistics, some of which the authors may have done, and it was just unclear, and others that may suggest changes to the models.

- The initial statistical analysis described in the methods says (lines 204-205) "A gaussian error distribution was assumed for every model..." yet the models described in the main text use count data and are stated as being Poisson models? Make it clearer which models are presented in the main text and which are presented in the supplement.
- There are times when certain numerical fixed effects in a model were z-transformed while others were not (for example, (lines 223-224 "To measure the effect of brooding duration, the explanatory variables were female brooding duration % (transformed to z-scores), number of female food deliveries to the chicks, and number of male food deliveries to the chicks."). To improve the stability of the models and to be able to directly compare the beta estimates of different fixed effects within the same model all numerical fixed effects should be z-transformed. This will put all the different measures on the same scale so effects can be more accurately estimated.
- The random effects of the models in the main text include year, species, nest ID, and time of observation. Is there a reason the authors did not nest "Nest ID" within "species"?
- For the results presented in the main text as well as in the supplement, the models include interaction terms. Typically, the main effects cannot be clearly interpreted while those same terms are included in an interaction term in the model, but the authors appear to do this. I do not think these rules would change for their quasi-Bayesian framework, but I could be mistaken. Were non-significant interactions removed, and the models rerun? How were main effects interpreted when interactions were significant? For example, how are the authors interpreting a "significant" main effects (such as brooding duration) when that term is also included in two different interaction terms in the same model, one of which is significant (table 2)?
- In the supplement, the authors describe several additional models that were run. This includes (lines 423-427) "... we fitted one linear mixed effect model for each of the six dependent behaviors (incubation duration %, # incubation events, male food deliveries to the female, brooding

duration, male food deliveries to the chicks, female food deliveries to the chicks).” Yet a linear mixed model is not appropriate for all these different response variables. For example, % time would be better modeled with a general linear mixed model with a beta distribution. If the number of incubation events per hour had to be square root transformed to use a linear model, would a different distribution be a better fit for the raw data?

- According to the methods, the abundance of flies was measured once when nests either failed or nestlings fledged (lines 191-200). Looking at figure S3, it becomes clear that these measurements were taken across the nestling period (nests failed at 2-11 days, or fledged? As there are fly measurements during all these points). Was whether a nest failed or fledged, and the timing of failure considered in the statistical analysis? Presumably nests that failed early would have less time to accumulate flies compared to nests that failed later or fledged? If nests were heavily parasitized, females may have abandoned nests, or reduced nest attendance and then abandoned- could this be driving the behavior patterns that you are seeing?

-I agree with the authors that parental age can have important impacts on behavior and that it should be examined if the data are available. Currently, the authors examine male age, but it is not mentioned why they do not examine female age (unless I missed something)? Because female behavior had the largest effect on parasites abundance, it seems that female age would possibly be more important in this case. Were the data unavailable?

-In the discussion, the authors predict strong selection on coordination of male and female behavior to enhance nest guarding. While they suggest that this is an area for future study, could they not test this with their data? If I understand things correctly, they looked at male and female behavior (male feeding, female brooding) and the interaction- but could they not take their data and look at the proportion of time that the nest was left unattended to look at coordinated nest guarding by both parents? For example, if males were visiting the nest to provision offspring when the female was absent?

Minor Comments:

-The authors do point out in the introduction that much work has been done on the role of parental provisioning and food compensation with avian nest parasites, and that the role of nest-attendance as a form of nest guarding against parasites is a fairly novel area. But I found the sentence in the abstract: (Lines 42-42) “Such predictions have been robustly tested in predation risk contexts, but little is known about parental care investment trade-offs under conditions of parasitism.” to be inaccurate. Nest parasite studies and manipulations have been done for decades and have become a standard model for testing the tradeoffs of parental care, parent-offspring conflict, and eco-immunology.

-I am curious if there is evidence for nest guarding or the role of nest attendance with preventing egg laying by avian brood parasites. If so, this could be a useful parallel to draw in the discussion. The authors do bring this up briefly with regards to cooperative breeding, but I think there may be more there.

-While the authors make a fairly good case that female nest-attendance is likely reducing egg laying by flies and thus reducing the abundance of parasites in the nest, this study is correlational, and this hypothesis remains to be thoroughly tested. It could be that females are just investing less in heavily parasitized nests. Given this, I would suggest changing the title slightly to more appropriately fit their study.

-The authors often refer to “male food delivery” without clarifying whether this was to females or to nestlings. Since they measured both, I think it is important to clarify throughout to avoid confusion.

-At times, particularly in the discussion, the writing can be a bit hard to follow. I think ideas could be better connected and flow more clearly within paragraphs.

-I am not familiar with what minimum longevity mean? Is this not just average adult lifespan?

-Lines 262-263. It seems that some numbers may have been flipped here or I am confused. "But females with longer brooding duration had, on average, 3.8 x fewer *P. downsi* (mean number of *P. downsi* [95% CrI] = 11.8 [7.3; 19.2]) than females with shorter brooding duration (mean number of *P. downsi* [95% CrI] = 3.1 [1.8; 5.2] | Fig. 1A, Table 2)" 11.8 is bigger than 3.1?

-Lines 328: missing "to"

-Is it possible to include raw data points in the main text figures? This would be more transparent and informative.

Author's Response to Decision Letter for (RSPB-2021-1668.R0)

See Appendix A.

Decision letter (RSPB-2021-1668.R1)

29-Oct-2021

Dear Dr Kleindorfer:

Your manuscript has now been peer reviewed and the reviews have been assessed by an Associate Editor. The reviewers' comments (not including confidential comments to the Editor) and the comments from the Associate Editor are included at the end of this email for your reference. As you will see, the reviewers and the Editors have raised some concerns with your manuscript and we would like to invite you to revise your manuscript to address them.

Research ethics:

Use of animals and field studies:

It is a condition of publication that you make available the data and research materials supporting the results in the article (<https://royalsociety.org/journals/authors/author-guidelines/#data>). Datasets should be deposited in an appropriate publicly available repository and details of the associated accession number, link or DOI to the datasets must be included in the Data Accessibility section of the article (<https://royalsociety.org/journals/ethics-policies/data-sharing-mining/>). Reference(s) to datasets should also be included in the reference list of the article with DOIs (where available).

Please submit a copy of your revised paper within three weeks. If we do not hear from you within this time your manuscript will be rejected. If you are unable to meet this deadline please let us know as soon as possible, as we may be able to grant a short extension.

Best wishes,
 Dr Sasha Dall
 Editor, Proceedings B
 mailto: proceedingsb@royalsociety.org

Associate Editor

Comments to Author:

I think this is an interesting paper on an important topic that is worth publishing in PRSB.

However I am not satisfied with the author's response to the reviewers comments. Of about ~20 comments between the two reviewers, ~ half were not adequately addressed in the manuscript itself. The two reviewers are clearly knowledgeable about this field, and it is safe to assume they are at least as knowledgeable as the average future reader of this paper. It is also safe to assume that they read the paper as or more carefully than most readers will. Thus if they raise a point of confusion, it should be taken as an opportunity to clarify that point in the manuscript for future readers. I would like the authors to take a second look at the reviewer comments, and make sure that rather than simply providing an explanation to reviewers in the response letter, they have provided that explanation or clarification to future readers in the manuscript.

Specifically:

- R1-comment1: The alternative hypothesis laid out by R1 deserves more than one sentence in the discussion that does not provide any information on what this hypothesis really is (eg unclear from added sentence what the mechanism behind the hypothesis is). I would like to see at least one sentence explaining the hypothesis with at least 1 or 2 references, and the one elaborating on how the supplementary analysis disproves it

- R1-3 (brooding duration): While the authors resolved the wording discrepancy regarding brooding duration in the abstract, it is still unclear what brooding duration is – please add a very brief elaboration to the Abstract

- R1-4: add the elaboration re brooding duration to the text

- R2-2 (effect of year): add the critical points to the MS and SI. Also elaborate there or in response why modelled species as a random effect (only 4 levels which is borderline for a random intercept) rather than including it as a fixed effect, or even testing for a year x species interaction

- R2-3 (analysing count data) – not clear from response – are count data being analysed in linear models (rather than Poisson generalized linear models)? Count data break assumptions of Gaussian models by definition, so the explanation provided is not enough to explain/justify this approach

- R2-6: (interpreting main effects when they are evolved in significant interactions) I don't think this response goes far enough. First, interactions need to be flagged right away when presenting results so that there is no opportunity for confusion by readers who are skimming. E.g. "Brooding duration had a negative effect on the number of P. downsi when all co-variates are at their mean (Fig. 1A, Table 2), but this effect depended on XYZ". (ie introduce interactions in same sentence where make claim about fixed effect).

Elaborate in text on "This effect depends significantly on male provisioning to chicks" – when is the effect apparent and when is it not.

Second, interactions also need to be flagged and explained in the Figures – a reader who is skimming (as most readers do) will see the strong line in Fig 1a (which is made stronger by not beginning the yaxis at zero) and not necessarily realize that this effect depends on two other factors (as depicted in Fig1b and Fig2). The fig caption needs to clarify this in the explanation of panel Fig1A. Fig A Y axis should start at zero.

Third, this nuance needs to be clear in the Abstract as well.

R2-7: "this is a common practice" is not a statistical justification. Please add a reference to the modern (ie last 10 years) statistical literature that backs up the use of square-root transformations in the pseudo-Bayesian framework the authors use

R2-8 (measuring fly abundance) – please add this clarification to the manuscript

R2-9 (parental age) –consider whether wording could be clarified either in methods where describe ageing via plumage is described or adding a reminder later in text re why female age no considered. (e.g. L165 & 167 – changing ‘and’ to ‘whereas’).

R2-MinorComments2 – I think the reviewer means avian brood parasites (e.g. cuckoos) here, which is not captured in the quoted sentence.

R2-MC5 – add explanation of minimum longevity to text. Clearly some readers are not familiar with the term and defining it will not take much additional text.

R2-MC8 – Reviewer comment was specifically about the main text figures, so author response about supplementary figures does not address it.

Additional minor comments:

L105 – clarify what ‘most’ means here and add few words to explain the variation in the mortality rates – if mortality rates are sometimes 20% it seems that most does not mean ‘most Darwin’s finch chicks’ . Does variation in 20-100% come from different years? Finch species? Islands?

Same paragraph – define nare or naris for general audience

L102/3 – clarify where larvae are oviposited (one would assume it was on chicks but seems like cant be if are present in eggs) – will help set up last part of paragraph

Author's Response to Decision Letter for (RSPB-2021-1668.R1)

See Appendix B.

Decision letter (RSPB-2021-1668.R2)

19-Nov-2021

Dear Dr Kleindorfer

I am pleased to inform you that your Review manuscript RSPB-2021-1668.R2 entitled "Female in-nest attendance predicts the number of ectoparasites in Darwin’s finch species" has been accepted for publication in Proceedings B.

The referee(s) do not recommend any further changes. However, please add your Dryad DOI to the Data Accessibility section.

Because the schedule for publication is very tight, it is a condition of publication that you submit the revised version of your manuscript within 7 days. If you do not think you will be able to meet this date please let me know immediately.

To upload your manuscript, log into <http://mc.manuscriptcentral.com/prsb> and enter your Author Centre, where you will find your manuscript title listed under "Manuscripts with Decisions." Under "Actions," click on "Create a Revision." Your manuscript number has been appended to denote a revision.

You will be unable to make your revisions on the originally submitted version of the manuscript. Instead, upload a new version through your Author Centre.

- 1) A text file of the manuscript (doc, txt, rtf or tex), including the references, tables (including captions) and figure captions. Please remove any tracked changes from the text before submission. PDF files are not an accepted format for the "Main Document".
- 2) A separate electronic file of each figure (tiff, EPS or print-quality PDF preferred). The format should be produced directly from original creation package, or original software format. Please note that PowerPoint files are not accepted.
- 3) Electronic supplementary material: this should be contained in a separate file from the main text and the file name should contain the author's name and journal name, e.g. `authorname_procb_ESM_figures.pdf`
All supplementary materials accompanying an accepted article will be treated as in their final form. They will be published alongside the paper on the journal website and posted on the online figshare repository. Files on figshare will be made available approximately one week before the accompanying article so that the supplementary material can be attributed a unique DOI. Please see: <https://royalsociety.org/journals/authors/author-guidelines/>

4) Data-Sharing and data citation

It is a condition of publication that data supporting your paper are made available. Data should be made available either in the electronic supplementary material or through an appropriate repository. Details of how to access data should be included in your paper. Please see <https://royalsociety.org/journals/ethics-policies/data-sharing-mining/> for more details.

If you wish to submit your data to Dryad (<http://datadryad.org/>) and have not already done so you can submit your data via this link <http://datadryad.org/submit?journalID=RSPB&manu=RSPB-2021-1668.R2> which will take you to your unique entry in the Dryad repository.

Once again, thank you for submitting your manuscript to Proceedings B and I look forward to receiving your final version. If you have any questions at all, please do not hesitate to get in touch.

Sincerely,
Dr Sasha Dall
Editor, Proceedings B
<mailto:proceedingsb@royalsociety.org>

Associate Editor Board Member

Comments to Author:

Thank you for taking care of the final edits. You were not wrong that there is a length limit, but the journal office will let you know if either the abstract or manuscript is over this limit.

Decision letter (RSPB-2021-1668.R3)

19-Nov-2021

Dear Dr Kleindorfer

I am pleased to inform you that your manuscript entitled "Female in-nest attendance predicts the number of ectoparasites in Darwin's finch species" has been accepted for publication in Proceedings B.

Data Accessibility section

Open Access

Paper charges

Sincerely,

Appendix A

Response to reviewer comments revision RSPB-2021-1668 “Female in-nest attendance predicts the number of ectoparasites in Darwin’s finch species”

Associate Editor

Board Member: 1

Comments to Author:

I have now received two reviews of your paper from reviewers who clearly read it carefully, both of whom provided useful feedback. Based on these reviews and my own assessment, I feel that the paper could be suitable for publication in PRSB, providing you can address the reviewers’ comments. In particular, I would pay attention to the following:

Reviewer 1’s comment that timing of breeding might in fact drive the observed correlation is worth considering – it is a logical alternative hypothesis and seems relatively easy to explore with the data at hand.

Our response: Yes, agreed. We have considered this alternative hypothesis and find no support. We include this line of reasoning and evidence in the Discussion and in Supplementary Tables 6 and 7.

Reviewer 2’s about the statistics are also important, particularly a) whether gaussian errors are appropriate for count data (usually not – alternatives include Poisson and negative binomial models), and b) being careful about interpreting main effects that are involved in significant interactions, and being clear about how interactions are treated (e.g. are non-significant interactions retained in or dropped from models)

Our response: Thank you – these comments were helpful to identify an error we had made, not in the distributions but in our explanation. Specifically, error distributions were correctly used but we have clarified explanations per model (e.g., Lines 214 to 219) including how main effects and interactions were interpreted.

Referee: 1

Overall, this is a very good study in terms of long-term data, content, analysis, and writing. The study is relevant regarding the urgent conservation needs for saving the unique Darwin finches in the Galapagos.

The main finding of the study is that broods with longer female brooding duration have fewer blood sucking parasitic fly larvae, and that the effect increases when males deliver more food (i.e. a main effect and a interaction effect being significant). The main proposed hypothesis for this finding is that female nest attendance deters parasitic flies from ovipositing in a host nest. While this is a non-experimental study where cause and consequence by definition remain unclear, there is some observational support for this hypothesis. Also the study is non-experimental, it is still important as a first demonstration of such a relationship between nest attendance and parasite numbers, and will clearly trigger many subsequent studies into the mechanisms and selection pressures responsible for the relationship.

Our response: Thank you for this overall positive assessment of the value of the study.

In my view, the main alternative hypothesis that should and can be evaluated with the available data is the following: In many bird species with resource-dependent territorial defence, the early breeding birds within the annual breeding season are the high-quality birds, plus early in the breeding season there are also much fewer parasites seeking hosts. Thus, the main finding here could be simply due to the co-variable given by the timing of breeding. I

think this needs to be included into the analysis, also for guiding subsequent studies. In addition to parasite pressure, the timing of bird breeding evolved to occur at times of highest abundance and quality of food. Hence, early in the annual breeding season, a female can attend nests for longer without a change in male food provisioning rates. Our response: This is an excellent suggestion. We used two approaches to assess the possible role of timing of nesting:

- 1) We included ‘nesting onset date’ (number of days since the beginning of the year) as a covariate. Unfortunately, the main model cannot sustain more variables as we are at the upper limit of covariates allowed by our sample size; when adding ‘breeding date’, the model did not converge. Therefore, we ran a reduced model removing all covariates that were not statistically significant in the main model:

*glmer(nr philornis~brooding duration*number of male feeds + breeding timing + (1/Year) + (1/MaleSpecies/Nest ID) + (1/time start), offset = obs_dur, family = poisson)*

Again, we found support for the result that brooding duration had a statistically meaningful effect on number of *P. downsi* parasites, while accounting for the breeding date.

- 2) We assessed whether nesting onset date explains significant variation in brooding duration or parasite burden (# *P. downsi* p[er nest). In neither case was their statistical support for an effect of breeding date on brooding behaviour or number of parasites.

Therefore, in summary, we found no support for the alternative hypothesis of an effect of breeding date on the patterns observed, which we now mention. In the first paragraph of the discussion we write: “In the supplementary material, we report lack of support for an alternative hypothesis that date of nesting onset explains female brooding duration and/or number of parasites in the nest (Tables S6 and S7).”

The abstract (and paper) needs some clarifications:

Food delivery: There is nowhere a description how food delivery has been measured, for example how, when, how long and how often. Furthermore, it is not stated what the variable expresses, hourly rate, daily rate, total rate over the brooding period, or total number of food items over the entire brooding period? The latter would be strongly influenced by the main significant variable in the study, i.e. brooding duration.

Our response: We added “per hour” to the methods, so the text now reads “(5) number of male visits per hour to the nest with food delivery to chicks; and 6) number of female visits per hour to the nest with food delivery to chicks.”

Brooding duration: The reader cannot figure out from the abstract, what the main significant variable exactly describes. In the methods only it is stated as “time female spends inside the nest during the chick phase (brooding duration, min per hr)”. This needs to be made clear in the abstract. In a following sentence of the abstract it says “female nest attendance duration” but it is not made clear whether this is the same as brooding duration.

Our response: Changed to read: “While the causal mechanisms remain unknown, we provide the first empirical study showing that female brooding duration is negatively related to the number of ectoparasites in the nest”

Prediction for selection for coordinated male and female nest attendance: In the Methods (line 154) it is stated that only females incubate eggs and hatchlings. It then remains unclear whether this prediction refers to just the hatchling period or to the entire nestling phase. It is further confusing because “brooding duration” refers to the entire chick (i.e. nestling) phase (see above).

Our response: Coordinated male and female behaviour at the nest could include male feeds to the female near the nest entrance (for example, at the end of an incubation bout to reduce time away from the nest by the female), or male food delivery to chicks that coincide with the end of female brooding events (the nest is guarded by the male while the female forages). We acknowledge there are different aspects pertaining to coordinated parental care across the nesting cycle and planning an analysis of these factors requires careful attention.

The Discussion of this paper has a strong evolutionary tack that Darwin would have liked.
Our response: Thank you.

Referee: 2

Overview: This manuscript examines the relationship between parental care and the abundance of an introduced nest parasite, the avian vampire fly, in three species of Galapagos finches. The study looked at several types of parental care across the nestling developmental period for both male and female parents: including incubation, brooding, and provisioning. The dataset of observations from 208 nests spans a 21-year period (11 years with data). The study finds that nests with longer female brooding durations during the early phase of nestling development (under 6 days old) had fewer flies. This effect was intensified in nests where there were more male food deliveries to nestlings. There were no significant relationships between other measures of parental care and the number of fly larva. The authors suggest that female brooding behavior during this key period likely limits the ability of adult flies to oviposit into the nests, thus limiting parasite abundance and that brooding and potentially coordination of male and female care in this case can be seen as a form of nest guarding. I appreciate the large sample size and many years of data presented in this study, which represent an impressive amount of field work, as well as the detailed look at parental care, which included different types of care for both males and females. This study is also novel as it examines the role of nest attendance as a parasite deterrent, while previous work has largely focused on the role of provisioning and food compensation with nest parasites (as the authors point out). I offer suggestions to improve the study as well as point out areas that need clarification below.

Our response: Thank you for this positive assessment about the value of the study to generate new ideas about parental care and parasitism.

Major Comments:

-I think more natural history information about the fly needs to come earlier in the paper. For readers who are unfamiliar with this system, that information is key to understand predictions for what types of parental care may be important to deter ovipositing and when during development that occurs. The authors should discuss when and how flies are targeting nests (what is known), the duration that flies target nests during the nestling developmental period. The authors do discuss this very briefly in the discussion (lines 358-361)- and for me, this made it much clearer why they were looking at incubation in addition to brooding and provisioning (initially, I assumed flies would only target nests with nestlings so was unsure

why they were looking at parental care during the egg phase). The fact that there have been measurable shifts through time in when flies are targeting nests and presumably the cost of flies (flies are getting smaller) is also relevant to this study and should be brought up earlier in the paper so readers can understand the results in context, especially because this dataset spans both islands and an extended period of time (21 years).

Our response: Thank you for pointing out this source of confusion. We have expanded the paragraph in the introduction to include more details on island differences in *P. downsi*. The oviposition behaviour changed after 2004 on Santa Cruz to fly oviposition during the host incubation phase. Otherwise, the pattern of smaller body size is similar across both islands. We added the following: “There are some island differences in *P. downsi* behaviour. For example, during 2000 to 2004 on Santa Cruz Island, *P. downsi* were only found in Darwin’s finch nests with chicks (100% prevalence), but since 2012 on Santa Cruz Island, *P. downsi* have regularly been found in Darwin’s finch nests with eggs (e.g. 80% prevalence) and in most nests with chicks (83% to 100% prevalence) 35. In contrast, on Floreana Island, *P. downsi* larvae and pupae are very uncommon in nests with eggs (2% prevalence) but occur in all highland nests with chicks (100% prevalence)36.”

-Building off the previous comment- I think it is interesting that there have been measurable shifts in fly behavior (timing of oviposition) and cost through time and wonder if the study could further explore that with regards to parental behavior. If I am understanding the results correctly, the *random effect of year, particularly for the brooding model presented in the main text is significant*. I would be curious if parental care is shifting through time, and if this nest guarding aspect of nest attendance is becoming stronger as the birds adapt to this parasite? This particularly interesting given the context that this is an introduced parasite.

Our response: This is a good suggestion. In our data we find no evidence that parental care of chicks (brooding duration, male food delivery, female food delivery) patterns changed across time. Here we attach our analyses for the reviewer. We made one linear mixed effect model for each of the following dependent variables: a) brood duration, b) male provisioning, and c) female provisioning. The explanatory variable was ‘Year of data collection’, and the random factors were (as in the main models) ‘Nest ID’ nested within ‘Species’ and the ‘Time of observation’. Below we attach the table with the coefficients for each of the models and three figures with a visualisation of the raw data:

	Brooding duration	Male food delivery	Female food delivery
Fixed effects		β (95% CrI)	
Intercept	16.56 (-31.4; 63.49)	28.98 (-16.92; 74.12)	-15.76 (-62.35; 30.06)
Year	-0.008 (-0.03; 0.02)	-0.01 (-0.04; 0.008)	0.007 (-0.02; 0.03)
Random effect		σ² (95%CrI)	
Species	0.00 (0.00; 0.00) *	0.00 (0.00; 0.00) *	0.17 (0.04; 0.52) *
Nest ID	0.65 (0.51; 0.85)	0.29 (0.22; 0.38)	0.47 (0.36; 0.61)
Time of observation	0.02 (0.01; 0.022)	0.30 (0.22; 0.39)	0.006 (0.003; 0.008)

* 0.00 indicate values < than 0.0001

Relationship between brooding behaviours and Year. The response variables Brooding duration, Male food delivery, and Female food delivery were z-transformed and modelled with a Gaussian error distribution (Linear mixed effect model). Estimates of fixed (β) and random (σ^2) parameters with their 95% credible intervals (CrI) are shown in brackets. Statistically meaningful effects are those where the CrI do not overlap zero (i.e. $posterior(p) > 95\%$) and are marked in bold.

Variation in brooding duration over time.

Variation in male food delivery (provisioning) over time.

Variation in female food delivery (provisioning) over time.

-I had several questions about the statistics, some of which the authors may have done, and it was just unclear, and others that may suggest changes to the models:

- The initial statistical analysis described in the methods says (lines 204-205) “A gaussian error distribution was assumed for every model...” yet the models described in the main text use count data and are stated as being Poisson models? Make it clearer which models are presented in the main text and which are presented in the supplement.

Our response: Thank you, yes, that was unclear. We have rephrased the sentence to improve clarity. Lines 214-216. We write: “For every statistical model (package ‘lme4’) the restricted maximum likelihood estimation method was applied, and all the assumptions were checked by visual inspection of the residual plots.”

- There are times when certain numerical fixed effects in a model were z-transformed while others were not (for example, (lines 223-224 “To measure the effect of brooding duration, the explanatory variables were female brooding duration % (transformed to z-scores), number of female food deliveries to the chicks, and number of male food deliveries to the chicks.”). To improve the stability of the models and to be able to directly compare the beta estimates of different fixed effects within the same model all numerical fixed effects should be z-transformed. This will put all the different measures on the same scale so effects can be more accurately estimated.

Our response: We agree and thank the reviewer for this observation. We have now redone the models with every dependent variable/covariate z transformed. We have i) clarified this in the methods (lines 234-236 and lines 238-241), ii) updated Tables 1 & 2, and iii) updated Figure 1.

- The random effects of the models in the main text include year, species, nest ID, and time of observation. Is there a reason the authors did not nest “Nest ID” within “species”?

Our response: As in the previous comment, we agree with the reviewer, this is a more elegant and probably more accurate approach. We have now included the nested random factors (Nest ID/Species) in our models and clarified this approach in the methods (Line 238 and Line 248).

- For the results presented in the main text as well as in the supplement, the models include interaction terms. Typically, the main effects cannot be clearly interpreted while those same terms are included in an interaction term in the model, but the authors appear to do this. I do not think these rules would change for their quasi-Bayesian framework, but I could be mistaken. Were non-significant interactions removed, and the models rerun? How were main effects interpreted when interactions were significant? For example, how are the authors interpreting a “significant” main effects (such as brooding duration) when that term is also included in two different interaction terms in the same model, one of which is significant (table 2)?

Our response: We agree with the reviewer that it can be difficult to interpret main effects when included in interactions. Implicit in the interpretation of linear models is the notion that effects and estimates can only be quantified while keeping the co-variates constant (any given value). We interpreted the main effect for the mean of our co-variates (which in our case is mathematically zero, as they are all z-transformed). We have now rephrased these results and stated them more accurately in Lines 273 to 280. We write: “During the chick feeding phase, female food delivery to chicks was not associated with the number of *P. downsi* in the nest (Table 2). Brooding duration had a negative effect on the number of *P. downsi* (when all co-

variates are at their mean; Fig. 1A, Table 2). This effect depends significantly on male provisioning to chicks (Fig. 1B, interaction term in Table 2). Also, this effect was attenuated with increasing age of the chicks and disappeared after the chicks were older than six days (Figure 2; interaction term Table S4), when females brooded less (Figure S3). We found the same relationship between brooding duration and *P. downsi* on both islands (Table S5, Figure S4, S5)."

When performing alternative models without non-significant interactions, the main effect remains robust. However, we decided to stick to the complete model as we think it is more likely to accurately reflect the system.

• In the supplement, the authors describe several additional models that were run. This includes (lines 423-427) "... we fitted one linear mixed effect model for each of the six dependent behaviors (incubation duration %, # incubation events, male food deliveries to the female, brooding duration, male food deliveries to the chicks, female food deliveries to the chicks)." Yet a linear mixed model is not appropriate for all these different response variables. For example, % time would be better modelled with a general linear mixed model with a beta distribution. If the number of incubation events per hour had to be square root transformed to use a linear model, would a different distribution be a better fit for the raw data?

Our response: We have checked and clarified the type of models applied to each variable. The reviewer is correct in that percentages are typically modelled with generalized linear mixed effect models (glmer) with beta distribution. We have now examined incubation time with an offset function for the observation duration. The results did not change. We have updated Figures S2 and S1, Tables S1 and S2 and clarified the methods description (e.g., Lines 432-446).

In relation to the square-root transformations, this is a very common practice. Although in all cases running the models without transformation resulted in already good fits (as assessed from the diagnostic residual plots), after transformation model fit was improved. We do not see a problem in applying this transformation to the data.

• According to the methods, the abundance of flies was measured once when nests either failed or nestlings fledged (lines 191-200). Looking at figure S3, it becomes clear that these measurements were taken across the nestling period (nests failed at 2-11 days, or fledged? As there are fly measurements during all these points). Was whether a nest failed or fledged, and the timing of failure considered in the statistical analysis? Presumably nests that failed early would have less time to accumulate flies compared to nests that failed later or fledged? If nests were heavily parasitized, females may have abandoned nests, or reduced nest attendance and then abandoned- could this be driving the behavior patterns that you are seeing?

Our response: We account for this issue in the brooding model that includes the variable 'chick age' (Figure 2, Table S4). 'Chick age' is equivalent to the chick age at which the nest failed during the chick feeding phase and is the day the nest was collected; therefore 'chick age' is a proxy for time exposed to potential parasitic events.

-I agree with the authors that parental age can have important impacts on behavior and that it should be examined if the data are available. Currently, the authors examine male age, but it is not mentioned why they do not examine female age (unless I missed something)? Because

female behavior had the largest effect on parasites abundance, it seems that female age would possibly be more important in this case. Were the data unavailable?

Our response: As stated in the methods (lines 158-163), only males become progressively black-headed (*Camarhynchus* spp.) or black-bodied (*Geospiza* spp.) with each annual moult until 5 years of age, allowing them to be aged between yearling to five years or older. Females remain olive green or olive grey across their lives and therefore age cannot be assessed in females using plumage.

-In the discussion, the authors predict strong selection on coordination of male and female behavior to enhance nest guarding. While they suggest that this is an area for future study, could they not test this with their data? If I understand things correctly, they looked at male and female behavior (male feeding, female brooding) and the interaction- but could they not take their data and look at the proportion of time that the nest was left unattended to look at coordinated nest guarding by both parents? For example, if males were visiting the nest to provision offspring when the female was absent?

Our response: Our proposed measures of parental coordination involve analysing male and female presence within 0-1m of the nest entrance, and the role of food provisioning to shorten time away from the nest, which was not included in our measures of food delivery to the nest or attendance inside the nest. The additional data preparation is possible but would take thought and time to plan properly.

Minor Comments:

-The authors do point out in the introduction that much work has been done on the role of parental provisioning and food compensation with avian nest parasites, and that the role of nest-attendance as a form of nest guarding against parasites is a fairly novel area. But I found the sentence in the abstract: (Lines 42-42) “Such predictions have been robustly tested in predation risk contexts, but little is known about parental care investment trade-offs under conditions of parasitism.” to be inaccurate. Nest parasite studies and manipulations have been done for decades and have become a standard model for testing the tradeoffs of parental care, parent-offspring conflict, and eco-immunology.

Our response: Yes, agreed that parental care trade-offs have been tested; we changed the sentence in the abstract to “less is known about alternative functions of parental care under conditions of parasitism”.

-I am curious if there is evidence for nest guarding or the role of nest attendance with preventing egg laying by avian brood parasites. If so, this could be a useful parallel to draw in the discussion. The authors do bring this up briefly with regards to cooperative breeding, but I think there may be more there.

Our response: Yes, this is an exciting avenue for future research. We mention this in the discussion (lines 360-362) when we write “One study suggests that cooperative breeding deters brood parasites from entering the nest to oviposit because of the additional nest guarding provided by additional helpers 76.”

-While the authors make a fairly good case that female nest-attendance is likely reducing egg laying by flies and thus reducing the abundance of parasites in the nest, this study is correlational, and this hypothesis remains to be thoroughly tested. It could be that females are just investing less in heavily parasitized nests. Given this, I would suggest changing the title slightly to more appropriately fit their study.

Our response: We agree and throughout have fully acknowledged the correlational nature of the study. The statistical models support our interpretation, as no other factor was associated

with parasite burden other than female brooding duration (and its effect was enhanced under conditions of high male food delivery). In line with your comment, we have also changed the title to “Female in-nest attendance predicts the number of ectoparasites in Darwin’s finch species”.

-The authors often refer to “male food delivery” without clarifying whether this was to females or to nestlings. Since they measured both, I think it is important to clarify throughout to avoid confusion.

Our response: ‘Male food delivery’ during the chick phase refers to male provisioning of chicks. We clarify ‘male food delivery to female’ during incubation versus ‘male food delivery to chicks’ during the chick feeding phase (clearly labelled in all tables)

-I am not familiar with what minimum longevity mean? Is this not just average adult lifespan?

Our response: ‘minimum longevity’ is a term used for assessing lifespan based on resighting or recapture data, which is what we used. The maximum age at recapture is the minimum longevity (because birds live for longer than the last recapture or resighting of a live bird). This term is standard in the literature but does take some getting used to.

-Lines 262-263. It seems that some numbers may have been flipped here or I am confused. “But females with longer brooding duration had, on average, 3.8 x fewer *P. downsi* (mean number of *P. downsi* [95% CrI] = 11.8 [7.3; 19.2]) than females with shorter brooding duration (mean number of *P. downsi* [95% CrI] = 3.1 [1.8; 5.2] | Fig. 1A, Table 2)” 11.8 is bigger than 3.1?

Our response: They were flipped, indeed. Thank you for picking this up. We have reformulated the reporting (Lines 274-277).

-Lines 328: missing “to”

Our response: Done.

-Is it possible to include raw data points in the main text figures? This would be more transparent and informative.

Our response: We have done so in the supplementary figures and include a new Figure S5 that also includes the incubation data.

Appendix B

Associate Editor

Comments to Author:

I think this is an interesting paper on an important topic that is worth publishing in PRSB. However I am not satisfied with the author's response to the reviewers comments. Of about ~20 comments between the two reviewers, ~ half were not adequately addressed in the manuscript itself. The two reviewers are clearly knowledgeable about this field, and it is safe to assume they are at least as knowledgeable as the average future reader of this paper. It is also safe to assume that they read the paper as or more carefully than most readers will. Thus if they raise a point of confusion, it should be taken as an opportunity to clarify that point in the manuscript for future readers. I would like the authors to take a second look at the reviewer comments, and make sure that rather than simply providing an explanation to reviewers in the response letter, they have provided that explanation or clarification to future readers in the manuscript.

Our response: Our intention was to be succinct, not disrespectful. We were under the impression that we were severely constrained with word count in the manuscript. We have now provided the requested clarifications in the manuscript. Given that response to reviewer comments is available online, we thought that our responses would be transparent to an interested and critical reader.

Specifically:

- R1-comment1: The alternative hypothesis laid out by R1 deserves more than one sentence in the discussion that does not provide any information on what this hypothesis really is (eg unclear from added sentence what the mechanism behind the hypothesis is). I would like to see at least one sentence explaining the hypothesis with at least 1 or 2 references, and the one elaborating on how the supplementary analysis disproves it.

Our response: We added this to the introduction lines 155-159: "Finally, we consider an alternative explanation, namely that date of nesting explains variation in parental care given changes in timing of activity associated with ambient temperature 39,40, invertebrate abundance 41 or other factors we did not measure. We also analysed nesting date in relation to number of *P. downsi* per nest as number of parasites may increase across the host nesting season .42"

In the results, we added lines 300-304: The effect of brooding on the number of *P. downsi* remained statistically meaningful while accounting for the date of nesting onset (Tables S6) and the date of nesting onset was not associated with female brooding duration and/or the number of parasites in the nest (Table S7). Finally, we found no evidence for shifts in parental behaviour (i.e., female brooding duration, male food deliveries, female food deliveries) across the years (Table S8).

In the discussion we write line 315: "There was no effect of nesting date on patterns of parental care or number of *P. downsi*, and no effect of parental care during the incubation phase on number of *P. downsi*."

In the supplementary material, we provide the description of the statistical models and the supporting tables.

- R1-3 (brooding duration): While the authors resolved the wording discrepancy regarding brooding duration in the abstract, it is still unclear what brooding duration is – please add a very brief elaboration to the Abstract

Our response: added line 56 “Nests with longer female brooding duration (mins per hr spent sitting on hatchlings to provide warmth) had fewer parasites...”

- R1-4: add the elaboration re brooding duration to the text

Our response: added line 210: “time female spends inside the nest sitting on hatchlings to provide warmth during the chick phase (brooding duration, min per hr)”

- R2-2 (effect of year): add the critical points to the MS and SI. Also elaborate there or in response why modelled species as a random effect (only 4 levels which is borderline for a random intercept) rather than including it as a fixed effect, or even testing for a year x species interaction.

Our response: We briefly report this (effect of year) in the results section. We also describe the statistical models and provide the supporting tables in the supplementary material. In the results lines 303-304 we state: “Finally, we found no evidence for shifts in parental behaviour (i.e., female brooding duration, male food deliveries, female food deliveries) across the years (Table S8)”.

We also briefly discuss the negative findings in the discussion (Lines 319 to 327):

“We found no effect of year on parental care variables in this study, which is perhaps surprising given strong natural selection by *P. downsi*. There are several possible reasons why female brooding duration did not get longer over time due to directional selection. Opposing environmental factors such as resource quality and abundance, thermal risk, and predation risk, which we did not measure, can potentially impact parental care decisions in a given year 3. Further, there is no evidence in birds that brooding duration or food delivery are heritable traits 61,62. Finally, given that Darwin’s finches can live to ~17 years, our generational sampling time window may be too shallow to detect such an evolutionary change should it occur, as the parents survive *P. downsi* but the offspring die 63”

In relation to the suggestion to include species as a fixed factor in this analysis, we added the following explanation in the description of the statistical models in the Supplementary Materials: “We did not consider the effect of species as fixed factor and its interaction with year for three reasons: i) not all species were sampled in all years, which generated large uncertainty, ii) when including species × year interaction, models did not converge, and iii) while not possible to include the interaction term for reasons just described, we were interested to test the general effect of year on parental care behaviours per nest.”

- R2-3 (analysing count data) – not clear from response – are count data being analysed in linear models (rather than Poisson generalized linear models)? Count data break assumptions of Gaussian models by definition, so the explanation provided is not enough to explain/justify this approach

Our response: We are cognizant of the application of Poisson generalized linear mixed models for count data, which is what we did throughout our analyses and have now clarified in the text. Lines 243 and 275 this is now explicitly stated.

- R2-6: (interpreting main effects when they are evolved in significant interactions) I don’t think this response goes far enough.

First, interactions need to be flagged right away when presenting results so that there is no opportunity for confusion by readers who are skimming. E.g. “Brooding duration had a negative effect on the number of *P. downsi* when all co-variates are at their mean (Fig. 1A, Table 2), but this effect depended on XYZ”. (ie introduce interactions in same sentence where make claim about fixed effect).

Elaborate in text on “This effect depends significantly on male provisioning to chicks” – when is the effect apparent and when is it not.

Our response: We have changed the text and rewritten lines 292-295 as: “Brooding duration had a negative effect on the number of *P. downsi* (when all co-variates are at their mean (Fig. 1A, Table 2), but this effect depended on male provisioning to chicks (interaction term in Table 2).”

Second, interactions also need to be flagged and explained in the Figures – a reader who is skimming (as most readers do) will see the strong line in Fig 1a (which is made stronger by not beginning the yaxis at zero) and not necessarily realize that this effect depends on two other factors (as depicted in Fig1b and Fig2). The fig caption needs to clarify this in the explanation of panel Fig1A. Fig A Y axis should start at zero.

Our response: This has been clarified in the new captions. In panels A, B and C from new figure 1 (with the raw data) the y-axis starts at zero.

Third, this nuance needs to be clear in the Abstract as well.

Our response: Clearly written in the abstract now as “this effect depended on male food delivery to chicks”.

R2-7: “this is a common practice” is not a statistical justification. Please add a reference to the modern (ie last 10 years) statistical literature that backs up the use of square-root transformations in the pseudo-Bayesian framework the authors use.

Our response: We think there is misunderstanding regarding the dependent variable in question. R2 pointed out: “If the number of incubation events per hour had to be square root transformed to use a linear model, would a different distribution be a better fit for the raw data?”. It is possible that “number of incubation events per hour” was interpreted as count data, however it is a continuous variable with a normal distribution (as it is a variable that depicts number of events relative to time). For this reason, it was first analysed with a Linear Mixed Effect Model where the fit was decent. Yet, we carried out a square-root transformation of the dependent variable as the model fit was much better. So, in essence, we carried out a square-root transformation of a continuous variable for a Linear Mixed Effect Model with a Gaussian error distribution. This is in fact common, irrespective of the statistical interpretation approach (frequentist or Bayesian). This is mentioned, for example, in Gelman and Hill ‘Data Analysis Using Regression and Multilevel/Hierarchical Models’ (2006). Here it is stated (in relation to linear models) “The square root is sometimes useful for compressing high values more mildly than is done by the logarithm.”. In addition, we could not find any references mentioning that such an approach is incorrect. If this would be the case, we would appreciate that the editor shares it and we could in turn present the supplementary results without transformation, as the output remains almost unchanged.

R2-8 (measuring fly abundance) – please add this clarification to the manuscript

Our response: This is detailed in a paragraph titled “Number of *Philornis downsi* per nest” lines 217-226.

R2-9 (parental age) –consider whether wording could be clarified either in methods where

describe ageing via plumage is described or adding a reminder later in text re why female age no considered. (e.g. L165 & 167 – changing ‘and’ to ‘whereas’).

Our response: Changed ‘and’ to ‘whereas’. Added sentence lines 176-178: “We only consider effects of male age on parental care because male age can be inferred from plumage colour whereas female age cannot be inferred from plumage colour.”

R2-MinorComments2 – I think the reviewer means avian brood parasites (e.g. cuckoos) here, which is not captured in the quoted sentence.

Our response: The reference we cite was about cuckoos: Canestrari, D., Marcos, J. M. & Baglione, V. Cooperative breeding in carrion crows reduces the rate of brood parasitism by great spotted cuckoos. *Animal Behaviour* 77, 1337-1344 (2009).

We added three references and changed the text lines 386-389 to read: “Perhaps helper birds provide additional nest guarding that deters brood parasites from entering the nest to oviposit 81, a form of frontline defence 82-84. Future research could explore the role of nest attendance and constraints to parental care in relation to nest guarding against brood parasites and ectoparasites 85.”

- Feeney, W. E., Welbergen, J. A. & Langmore, N. E. The frontline of avian brood parasite–host coevolution. *Animal Behaviour* 84, 3-12 (2012).
- Medina, I. & Langmore, N. E. Batten down the hatches: front-line defences in an apparently defenceless cuckoo host. *Animal Behaviour* 112, 195-201 (2016).
- Noh, H.-J., Jacomb, F., Gloag, R. & Langmore, N. E. Frontline defences against cuckoo parasitism in the large-billed gerygones. *Animal Behaviour* 174, 51-61 (2021).

R2-MC5 – add explanation of minimum longevity to text. Clearly some readers are not familiar with the term and defining it will not take much additional text.

Our response: added to read as follows lines 180-181: “Minimum longevity (calculated as the age at first capture plus the number of years until the last recapture) in these finches is 12–17 years 44.”

R2-MC8 – Reviewer comment was specifically about the main text figures, so author response about supplementary figures does not address it.

Our response: We have now included a new Figure 1 showing the raw data.

Additional minor comments:

L105 – clarify what ‘most’ means here and add few words to explain the variation in the mortality rates – if mortality rates are sometimes 20% it seems that most does not mean ‘most Darwin’s finch chicks’. Does variation in 20-100% come from different years? Finch species? Islands?

Our response: Changed line 105 to “On average, 55% of chicks die....” Which was calculated from the cited review analysing mortality caused by *P. downsi* in Darwin’s finch hosts across 31 studies.

Same paragraph – define nare or naris for general audience

L102/3 – clarify where larvae are oviposited (one would assume it was on chicks but seems like cant be if are present in eggs) – will help set up last part of paragraph

Our response: Rewritten lines 109-114 as “and adult finches that survived parasitism as chicks often sustain enlarged or malformed nares (nasal openings) as adults 33,34. *Philornis downsi* females oviposit eggs onto nesting material 23 and perhaps onto chicks; after hatching, larvae crawl inside the nares of the chicks to consume blood or keratin 11. Increasingly over the past decade, larvae on Santa Cruz Island are suspected to consume the blood of incubating females 35, with observations of temporal and island differences in *P. downsi* behaviour.”